# Transcriptome Analysis Revealed the Dynamic and Rapid Transcriptional Reprogramming Involved in Cold Stress and Related Core Genes in the Rice Seedling Stage

**DOI:** 10.3390/ijms24031914

**Published:** 2023-01-18

**Authors:** Bian Wu, Siyuan Chen, Shiyuan Cheng, Changyan Li, Sanhe Li, Junxiao Chen, Wenjun Zha, Kai Liu, Huashan Xu, Peide Li, Shaojie Shi, Guocai Yang, Zhijun Chen, Kai Liu, Aiqing You, Lei Zhou

**Affiliations:** 1Laboratory of Crop Molecular Breeding, Ministry of Agriculture and Rural Affairs, Hubei Key Laboratory of Food Crop Germplasm and Genetic Improvement, Food Crops Institute, Hubei Academy of Agricultural Sciences, Wuhan 430070, China; 2Hubei Hongshan Laboratory, Wuhan 430070, China; 3College of Life Sciences, Wuhan University, Wuhan 430072, China; 4National Key Laboratory of Crop Genetic Improvement, Huazhong Agricultural University, Wuhan 430070, China

**Keywords:** rice, cold stress, transcriptome, DEG, genome re-sequencing

## Abstract

Cold damage is one of the most important environmental factors influencing crop growth, development, and production. In this study, we generated a pair of near-isogenic lines (NILs), Towada and ZL31, and Towada showed more cold sensitivity than ZL31 in the rice seedling stage. To explore the transcriptional regulation mechanism and the reason for phenotypic divergence of the two lines in response to cold stress, an in-depth comparative transcriptome study under cold stress was carried out. Our analysis uncovered that rapid and high-amplitude transcriptional reprogramming occurred in the early stage of cold treatment. GO enrichment and KEGG pathway analysis indicated that genes of the response to stress, environmental adaptation, signal transduction, metabolism, photosynthesis, and the MAPK signaling pathway might form the main part of the engine for transcriptional reprogramming in response to cold stress. Furthermore, we identified four core genes, *OsWRKY24*, *OsCAT2*, *OsJAZ9,* and *OsRR6,* that were potential candidates affecting the cold sensitivity of Towada and ZL31. Genome re-sequencing analysis between the two lines revealed that only *OsWRKY24* contained sequence variations which may change its transcript abundance. Our study not only provides novel insights into the cold-related transcriptional reprogramming process, but also highlights the potential candidates involved in cold stress.

## 1. Introduction

Rice (*Oryza sativa* L.) is one of the major cereal crops in the world and plays a key role in global food security [1]. Cold damage as a kind of natural disaster is often encountered during the whole growth and development of rice, and limits its growth, geographical distribution, yield, and quality [2,3,4]. Especially for rice seedlings, low temperature causes damage to leaves and roots, a withered state, and even death [5]. In addition, some cold-sensitive rice varieties, as well as early season and direct-seeding rice, are more likely to suffer from continuous cold damage, resulting in a large loss of rice seedlings. Therefore, it is an unavoidable challenge to study the internal dynamic changes and regulatory mechanisms in response to cold stress, which is of great scientific significance for the cultivation of cold-resistant varieties and the improvement of yield in rice.

In the process of long-term natural evolution and artificial domestication, rice has developed complex strategies for rapidly sensing and effectively responding to cold stress. Cold is a physical signal which is firstly perceived by putative cold sensors, including cellular membranes, calcium (Ca^2+^) channels, and G-protein regulators [6]. CHILLING-TOLERANCE DIVERGENCE1 (COLD1), a regulator of G-protein signaling, was reported to interact with the G-protein *α* subunit to activate the Ca^2+^ channel for sensing low temperature, but further details on how COLD1 senses cold stress remain unknown [2]. After the signal reception, a sophisticated cold stress-specific signal transduction pathway is rapidly initiated, involving Ca^2+^, reactive oxygen species (ROS), plant hormone, and mitogen-activated protein kinase (MAPK) signal transduction [7]. Previous studies have shown that CYCLIC NUCLEOTIDE-GATED Ca^2+^ channels (CNGCs), including OsCNGC9, 14, and 16, induced by cold stress, mediated Ca^2+^ signaling and enhanced cold tolerance in rice [8,9]. ROS as secondary messengers triggered cold stress-related responses, while elevating the transcript abundance of ROS scavenging-related genes such as peroxidase could enhance cold tolerance [10]. HAN1 was reported to be involved in catalyzing the conversion of active jasmonic acid (JA) to the inactive form and modulated JA-mediated cold tolerance in temperate japonica rice [5]. MPK3 and MPK6 were rapidly activated after cold treatment, thus negatively regulating the cold response, whereas MPK4 positively regulated the cold response by suppressing MPK3 and MPK6 activity [11]. At present, although some signaling pathways of cold stress have been revealed, there are still many details and new approaches waiting to be uncovered.

Among the reported cold stress-related genes in rice, a large number of them belong to transcription factors (TFs), which form a complex regulatory network to instantly respond to alterations of ambient temperature by influencing the expression of downstream genes. *OsMYB30*, a cold-responsive R2R3-type MYB gene, increased cold sensitivity by downregulating *β*-amylase genes [12]. *OsJAZ9* is a member of the TIFY TF family and is upregulated under cold stress, while OsJAZ9 interacts with OsMYB30 to repress *β*-amylase expression and cold tolerance [12,13]. Population genetics studies identified OsbZIP73 as a positive regulator of cold tolerance via interacting with OsbZIP71 to modulate ABA levels and ROS homeostasis [14]. *OsDREB1A* and *OsDREB1B,* belonging to the AP2/ERF TF family, were both induced by cold, while improving their expression in transgenic *Arabidopsis* and rice significantly increased cold tolerance [15,16,17]. OsWRKY63 negatively regulated cold tolerance through the OsWRKY63–OsWRKY76–OsDREB1B transcriptional regulatory cascade in rice [10]. TFs play important roles in response to cold stress, while the complicated relationships among them in interaction and regulation need to be further disentangled.

To gain a new understanding of cold-related pathways and genes, in this study, a pair of near-isogenic lines (NILs) showing different cold sensitivity in the rice seedling stage were used as the experimental material, and their phenotypes of survival rates were compared after 4 °C cold treatment. RNA-seq was used to analyze the dynamic transcriptome changes of rice seedlings of NILs by using samples at multiple time points under cold treatment. We further performed the sequence alignment of the candidate genes through genome re-sequencing. The study reveals the regulatory pathways involved in cold stress, and predicts the cold-related genes of rice, laying a foundation for the next functional verification.

## 2. Results

### 2.1. Rice ZL31 Exhibits Strong Cold Sensitivity in the Seedling Stage

In a previous study [18], we constructed NILs derived from backcrossing and self-interbreeding using parents of the cold-tolerant donor Kunmingxiaobaigu (KMXBG) and the relative cold-sensitive Japanese commercial *japonica* cv. Towada. Under 4 °C cold treatment, we screened a NIL named ZL31, which was more sensitive to cold than Towada in the seedling stage (Figure 1A). Compared with Towada, the seedling survival rate of ZL31 decreased from 68.7% to 26.7% (Figure 1B). It is speculated that their different manifestation towards cold may be related to the divergence of gene expression patterns.

### 2.2. Dynamic Transcriptome Changes in Different Cold Treatment Times

To investigate the gene expression associated with the cold sensitivity in ZL31, in-depth comprehensive transcriptome profiles were compared with Towada through RNA-seq experiments. The rice seedlings were treated at 4 °C for 0, 3, and 12 h, and then leaf samples with three biological replicates were collected for RNA-seq. In total, 18 samples were subjected to RNA-seq, resulting in a range of 1.94–3.09 million 100-bp paired-end reads per sample.

To visualize the variation as well as the similarity for all samples, we performed a principal component analysis (PCA) on the fragments per kilobase of transcript per million mapped reads (FPKM) of all the detected genes. The PCA plot showed 40% and 44% variance among ZL31 vs. Towada samples, respectively, and the data for three biological replicates were clustered closely and were separated by the time point and genotypes (Figure 2A). Using the DESeq2 comparison of all groups, we identified differential expression genes (DEGs) (|Log_2_FoldChange| > 1, *p* < 0.01) between the cold treatment and control in Towada and ZL31 at each time point. A large number of DEGs in Towada were identified, especially at 12 h (1056 upregulated and 675 downregulated). Cold treatment induced more dramatic transcriptional changes in ZL31 (1001 upregulated and 331 downregulated) than in Towada (651 upregulated and 197 downregulated) at 3 h (Figure 2B). In total, 2098 and 2490 DEGs, accounting for approximately 5% of the rice genes, were identified in Towada and ZL31, respectively. Among these DEGs, 1386 were commonly regulated in both cultivars (Figure 2C). Together, the data indicated that the cold condition induced dramatic and dynamic transcriptional regulation in rice. Remarkably, ZL31 mounted a faster and stronger transcriptional response during the early stage of the cold treatment than Towada, which may be related to ZL31 being more cold-sensitive.

### 2.3. The Basal Expression of Genes in Towada Differs from That in ZL31

To figure out which genes were differentially expressed at the basal level in ZL31 compared with Towada before cold treatment, we analyzed DEGs of the samples at 0 h. There were 148 DEGs in ZL31 vs. Towada at 0 h (67 downregulated and 81 upregulated) (Figure 3A,B). We then functionally analyzed the 148 DEGs with respect to their Gene Ontology (GO) enrichment and Kyoto Encyclopedia of Genes and Genomes (KEGG) pathways to understand the biological relevance underlying these genes. GO enrichment analysis showed that the most enriched genes participated in “biological process”, followed by “cellular component” and “molecular function”. Among these categories, “photosynthesis”, “thylakoid”, and “metabolites and energy” were the most significantly enriched (Figure 3C). Consistently, photosynthesis- and metabolism-related terms, “photosynthesis proteins” and “energy metabolism”, etc., were also significantly enriched in KEGG pathway enrichment analysis (Figure 3D). These results illustrated that photosynthesis- and metabolism-related genes were differently regulated in ZL31 and Towada, which might be part of the reason for the difference in the cold sensitivity of the different varieties.

### 2.4. Rapid and Dramatic Transcriptional Reprogramming Occurs in the Early Stage of Cold Treatment

To gain more details about the alterations of transcriptional levels in Towada and ZL31 under cold treatment, we compared DEGs’ overlaps between them at each time point. A large number of common genes had changed, such as 583 DEGs (85 downregulated and 498 upregulated) at 3 h and 1009 DEGs (360 downregulated and 649 upregulated) at 12 h (Appendix A). We further overlapped and found that a total of 271 DEGs were involved in the response to cold stimulus (Figure 4A, Appendix A). Based on the FPKM of these DEGs, hierarchical clustering was performed and a global view of the expression levels at each time point was generated in Towada and ZL31. As shown in Figure 4B, a large portion of DEGs was upregulated and induced more strongly at 3 h in ZL31 and at 12 h in Towada, suggesting that a rapid and dramatic transcriptional reprogramming occurs in response to cold stimulus in the early stage.

GO enrichment and KEGG pathway analysis were performed to determine the functional classification and pathway assignment of the 271 DEGs. Remarkably, GO terms related to stress, including “response to stimulus”, “response to abiotic stimulus”, “response to stress”, “response to endogenous stimulus”, and “photosynthesis”, etc., were significantly enriched in the GO analysis (Figure 4B). In addition to metabolism- and photosynthesis-related pathways, DEGs mainly belonged to KEGG terms, including “environmental adaptation”, “environmental information processing”, “signal transduction”, “plant–pathogen interaction”, “plant hormone signal transduction”, and “MAPK signaling pathway”, etc. (Figure 4C). Together, these results indicate that the genes of the response to stress, environmental adaptation, signal transduction, metabolism, photosynthesis, and the MAPK signaling pathway are involved in the response to cold stress. Thus, we hypothesized that these DEGs might form the main part of the engine for transcriptional reprogramming in response to cold stress.

### 2.5. Identification of Core Genes Related to Differential Cold Sensitivity between Towada and ZL31

To identify which genes played key roles in the difference of the response to cold stress between Towada and ZL31, we performed Venn diagram analysis of the DEGs from Towada and ZL31 at all time points. The diagrams showed that there were 40 genes with markedly different expression levels between the two varieties under cold treatment of 3 or 12 h (Figure 5A, Appendix A). Hierarchical clustering of the FPKM values of those 40 genes showed that most DEGs were upregulated under cold treatment and their expression levels were significantly higher in ZL31 than in Towada (Figure 5B). These results imply that ZL31 is more sensitive to cold in the seedling stage, which may be related to the differential expression of these genes among different varieties.

Next, GO analysis indicated that the DEGs mainly belonged to “response to abiotic stimulus”, “response to stimulus”, “response to stress”, “catabolic process”, and “response to endogenous stimulus”, etc. (Figure 5C). The KEGG pathway analysis showed that “signal transduction”, “environmental information processing”, “polyketide biosynthesis proteins”, “MAPK signaling pathway”, and “plant hormone signal transduction” were most significantly enriched (Figure 5D). To find the key genes affecting the divergence of cold sensitivity in the two varieties, we further focused on the DEGs of these pathways. As a result, there were four upregulated genes that might be involved in regulating the divergence, including *OsWRKY24* (LOC_Os01g61080), *OsCAT2* (LOC_Os02g02400), *OsJAZ9* (LOC_Os03g08310), and *OsRR6* (LOC_Os04g57720). Among the genes, the highest transcription abundance was different: *OsWRKY24* and *OsJAZ9* at 3 h, while *OsCAT2* and *OsRR6* at 12 h under cold treatment. At these key time points, all four genes showed a consistent expression trend, and their transcription levels in ZL31 were significantly higher than those in Towada (Table 1). Moreover, we performed qRT-PCR to verify the above results and obtained a consistent conclusion (Figure 5E). The previous study showed that the four genes had already been cloned and reported to be related to hormone signal transduction, such as GA, ABA, JA, and CK, respectively [19,20,21,22,23]. Among them, OsJAZ9, OsCAT2, and OsRR6 were reported to participate in the response to abiotic stresses, including drought, salinity, and low temperatures [12,13,21,24]. Taken together, these analyses indicated that the above four genes are potential candidates affecting cold sensitivity.

### 2.6. Sequence Alignment of Candidate Genes between Towada and ZL31


We further compared the sequence variations of these candidate genes, including the 2 kb promoter and the coding sequence, between Towada and ZL31. The result showed that only *OsWRKY24* (LOC_Os01g61080) contained sequence variations, which occurred in the promoter, 5′ untranslated region (UTR), and the exon (Table 2). Moreover, nine polymorphisms were detected in the regulatory region, including two insertion/deletions (InDels) and seven single-nucleotide polymorphisms (SNPs), leading to changes in TF binding sites, such as MYB, etc. (Appendix A). Thus, we speculate that these variations may cause the divergence of transcript abundance of *OsWRKY24* in the two varieties, and functional characterization will be verified in future studies.

## 3. Discussion

Cold stress as a major environmental factor severely limits the growth and development, as well as the improvement of yield and quality, in rice. According to the degree of cold stress, the plant undergoes corresponding changes at the phenotypic, physiological, and molecular levels to respond to the low temperature, including photosynthetic rate, redox homeostasis, hormone signal transduction, transcriptional regulation, and post-translational modification [4,7]. In this study, we integrated the transcriptomic analysis and genome data to reveal the rapid and dramatic transcriptional reprogramming process in the early stage of cold treatment and predict the cold sensitivity-related candidate genes.

It has long been known that a host of alterations occur in gene expression when plants are subjected to cold stress [25,26,27,28]. As reported, both up- and down-regulation of gene expression occur, but generally more genes are upregulated than downregulated [29]. In our study, we also found that the number of upregulated DEGs was higher than that of downregulated DEGs in Towada or ZL31 at 3 or 12 h of cold treatment (Figure 2B and Figure 4B). It suggests that rice seedlings may need to sense, transmit, and respond to the cold signal by inducing gene expression, ultimately adapting to changes in environmental temperature. Meanwhile, plant materials with different genetic backgrounds had diverse responses to cold stress [4,29]. Our results showed that upregulated genes at 3 h of cold treatment in ZL31 were significantly more than in Towada (Figure 2B), which may be related to their differences in cold sensitivity. Previous studies revealed that the pathways involved in cold stress mainly included Ca^2+^ signaling, ROS homeostasis, plant hormone, and MAPK signal transduction [7]. Liu et al. [4] carried out a comprehensive analysis of the rice transcriptome and lipidome and confirmed that the fluidity and integrity of the photosynthetic membrane under cold stress led to the reduction of photosynthetic capacity, and lipid metabolism, including membrane lipid and fatty acid metabolism, might be an important factor in rice cold tolerance. Our transcriptome analysis has also obtained consistent results and indicates that the genes of response to stress, environmental adaptation, plant hormone signal transduction, metabolism, photosynthesis, and the MAPK signaling pathway are involved in the response to cold stress (Figure 4B,C). Among metabolism pathways, “glycerophospholipid metabolism” and “lipid metabolism” were significantly enriched (Figure 4C), verifying the importance of membrane lipid remodeling for rice adaptation to cold stress.

At present, 107 cloned genes have been reported to be involved in cold stress in rice, and about 34% of them belong to transcription factors such as WRKY, MYB, AP2, NAC, bHLH, bZIP, TCP, MADS, and Zinc finger protein (http://www.ricedata.cn/index.htm, accessed on 19 December 2022). Yokotani et al. [30] discovered that *OsWRKY76,* as a positive regulator factor, facilitated cold tolerance, and overexpression of *OsWRKY76* in rice plants resulted in drastically improved tolerance to cold stress by leading to the increased expression of abiotic stress-associated genes such as peroxidase and lipid metabolism genes. Another WRKY TF, *OsWRKY71*, was significantly upregulated under cold stress, and the survival rate and photosynthetic capacity of transgenic overexpressed rice seedlings were markedly better than those of the control after 4 °C cold treatment, indicating that *OsWRKY71* played a positive role in cold tolerance [31]. Conversely, *OsWRKY63*-overexpressing and knockout rice lines were, respectively, more sensitive and tolerant to cold stress compared to the control, meaning that *OsWRKY63* negatively regulated cold tolerance [10]. The interaction of OsWRKY76 and OsbHLH148 synergistically promoted the expression of the cold-tolerant gene *OsDREB1B*, whereas OsWRKY63 directly inhibited the expression of *OsWRKY76* and mediated the OsWRKY63–OsWRKY76–OsDREB1B transcriptional regulatory cascade [10]. In this research, we predicted four core genes affecting cold sensitivity, and among them, there were two TFs, *OsWRKY24* and *OsJAZ9*, which were significantly induced under cold treatment and expressed higher in ZL31 than Towada (Table 1). *OsWRKY24* was cloned as a transcriptional repressor to inhibit both GA and ABA signaling in aleurone cells [19,20]. However, it has not been reported whether OsWRKY24 participates in cold sensitivity in the same way. Sequence alignment of *OsWRKY24* showed that many variations occurred in the promoter, 5′ UTR, and the exon (Table 2), which may be responsible for the divergence of transcript abundance of *OsWRKY24* and phenotype between Towada and ZL31. OsJAZ9, also named OsTIFY11a, of the TIFY family, was identified as a transcriptional regulator in JA signaling and modulates salt stress tolerance in rice [22]. Expression profiles of *OsTIFY* genes in seedling leaves showed that *OsJAZ9* was strongly induced by cold [13]. Additionally, OsJAZ9 directly interacted with OsMYB30, a positive regulator of cold sensitivity, to co-suppress the expression of *β*-amylase genes and reduce cold tolerance [12]. Nevertheless, we did not discover any sequence variations in *OsJAZ9* between Towada and ZL31 (Table 2), suggesting that the difference in expression levels may be caused by the regulation of other genes.

The response of plants to low temperature is a complex process, and the molecular mechanism and regulatory network are still elusive. This study expands the understanding of the dynamic transcriptional reprogramming program under cold stress and supplies an important reference for the future research on cold tolerance of rice and other cereal crops. Besides, our results provide a new insight for accelerating the identification of novel cold-related genes and revealing the possible mechanisms in rice.

## 4. Materials and Methods

### 4.1. Experimental Material

Two rice materials, Towada and ZL31, were used in this study. ZL31 was one of the BC_6_F_7_ NILs developed by backcrossing and self-interbreeding using parents of cold-tolerant KMXBG and relative cold-sensitive Towada. ZL31 was more sensitive to cold stress than Towada in the seedling stage, so it was screened for further research.

### 4.2. Evaluation of the Survival Rate of Rice Seedlings under Cold Treatment

To ensure complete dryness and break any dormancy, seeds were dried at 38 °C for 48 h. The seeds were soaked with 10% sodium hypochlorite for disinfection, and then washed with sterile water. Then, the seeds were germinated in dishes with wet filter papers at 30 °C for 7 days. A total of 30 uniformly germinating seedlings of each line were planted in a ceramic pot and grown to the 3–4 leaf stage at 28 °C/24 °C, after which they were moved to a 4 °C growth cabinet for 4 days. The survival rate of seedlings was counted after 7 days of recovery culture under normal conditions. Each line carried out three biological repeated experiments.

### 4.3. RNA Extraction and qRT-PCR

The 3–4 leaf stage seedings of Towada and ZL31 were cultivated in a 4 °C growth cabinet for 0, 3, and 12 h. The seedings were collected and frozen in liquid nitrogen and stored at −80 °C until use, and each sample included 3 replicates, with 10 plants per replicate. Total RNA was extracted using a Trizol reagent kit (Invitrogen, Carlsbad, CA, USA) according to the manufacturer’s protocol and then the qualified RNAs were used for transcriptome sequencing. Quantitative RT-PCR was performed on a QuantStudio6 Flex machine using SYBR Green PCR reagent according to the manufacturer’s instructions. All assays were performed with three biological and three technical replications. The rice *actin1* gene served as the internal control to normalize gene expression. All the primers used for RT-qPCR analysis are listed in Appendix A.

### 4.4. Transcriptome Sequencing Analysis

RNA-sequencing was performed using nanopore full-length sequencing by Biomarker Technologies (Beijing, China). The raw data were quality-checked using Fastp software, and then we removed the sequencing junction to obtain clean data using Trimmomatic software [32,33]. The clean reads were further used for assembly and mapped to the rice reference genome (MSU7.0) using HISAT2 software [34].

### 4.5. Identification of Differentially Expressed Genes

Gene expression was quantified by counting the number of reads mapped to each gene using featureCounts software [35]. DESeq2 was employed to estimate the fold change and differentially expressed genes from the read counts data of gene expression level, given in fragments per kilobase of exon per million mapped fragments (FPKM) [36]. The *p*-values were adjusted for multiple testing using the default method, integrated with iDEP0.951 [37]. The transcripts of DEGs were determined with the parameters using |log_2_FoldChange| > 1 and *p* < 0.01.

### 4.6. Gene Function Annotation

Gene ontology (GO) enrichment analysis was performed using the functions of the hypergeometric distribution test for the calculation of GO terms. All DEGs were mapped to GO terms in the Gene Ontology database (www.geneontology.org, accessed on 10 November 2022) and the number of genes was calculated for each term by using TBtools software. The Kyoto Encyclopedia of Genes and Genomes (KEGG) enrichment analysis was performed to retrieve the enriched pathway, using padj < 0.05 as a threshold for significantly enriched DEGs in TBtools software [38].

### 4.7. Genome Re-Sequencing and Annotation

Towada and ZL31 were sequenced using the illumine HiSeq2000 instrument and the raw sequencing results were uploaded to NCBI. Fastp and Trimmomatic software were employed to check the quality of the raw data and remove sequencing junctions to obtain clean data. Subsequently, the sequencing data were aligned to the rice reference genome (MSU7.0) using BWA software [39]. SAMtools and BCFtools were used to identify SNPs and InDels. Only alignments with mapping quality ≥ 40 were used for the alignment, and bases with base quality ≥ 10 were used to identify SNPs and InDels [40,41]. Only the reads which uniquely mapped to the genomic sequence were retained for further analysis. Finally, SNPs and InDels were annotated using SnpEff software [42].

## Figures and Tables

**Figure 1 ijms-24-01914-f001:**
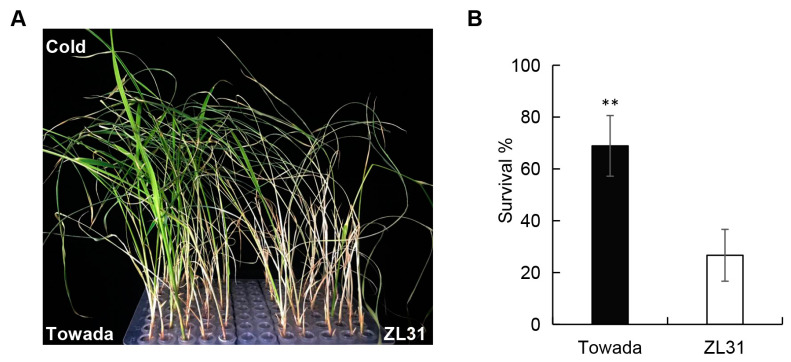
Identification of cold sensitivity of Towada and ZL31 in the seedling stage. (**A**) ZL31 was more sensitive to cold in the seedling stage under 4 °C cold treatment. (**B**) Comparative survival rate of seedlings of Towada and ZL31 under 4 °C cold treatment. A *t*-test was used to analyze the differences between means, presented as mean ± standard deviation, where ** shows that *p* < 0.01.

**Figure 2 ijms-24-01914-f002:**
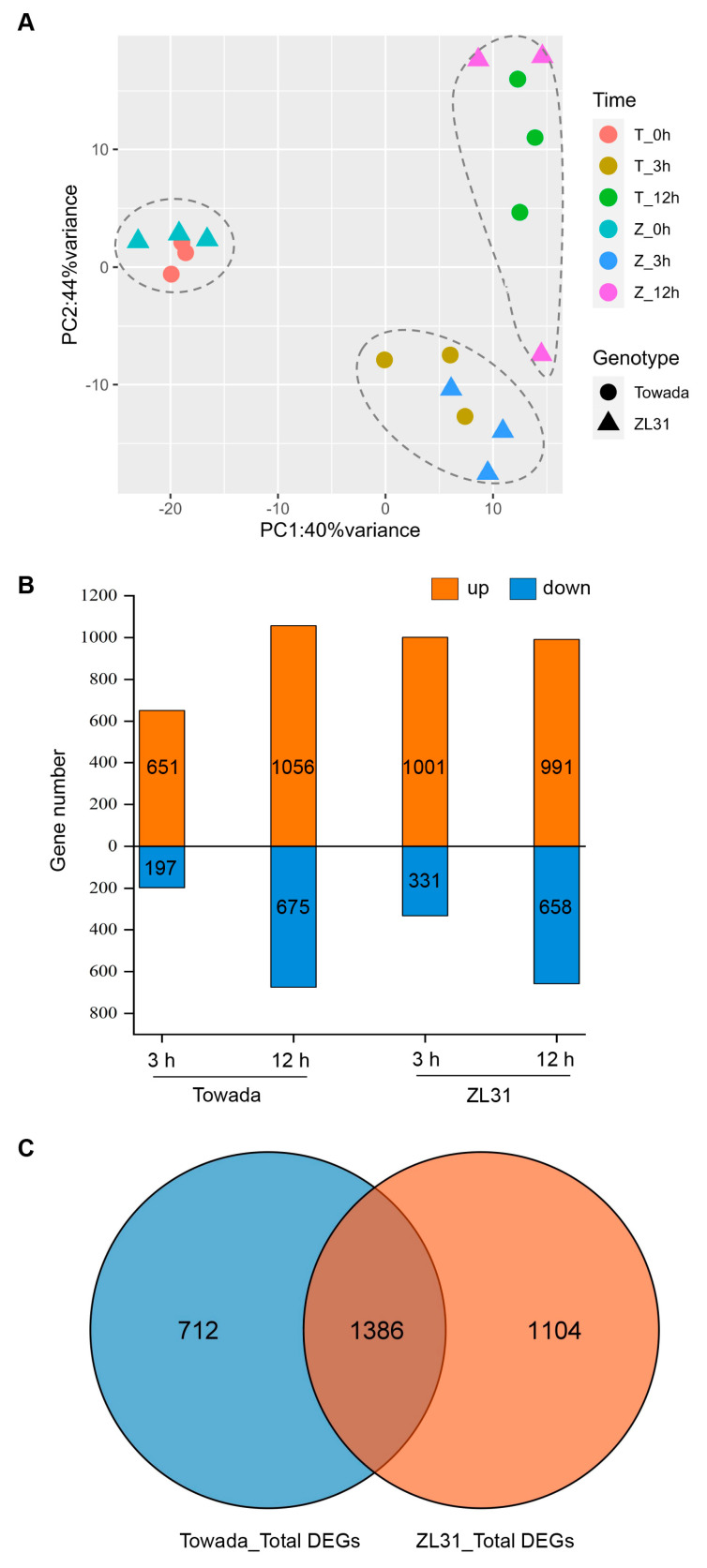
Overview of transcriptome data and differentially expressed genes (DEGs) in the rice response to cold stress. (**A**) Principal component analysis of the time−series transcriptome data in Towada and ZL31. (**B**) The numbers of up− and down−regulated genes in Towada and ZL31 at 3 and 12 h of cold treatment compared with the control (0 h) are shown. (**C**) Venn diagram of total DEGs in Towada compared with ZL31.

**Figure 3 ijms-24-01914-f003:**
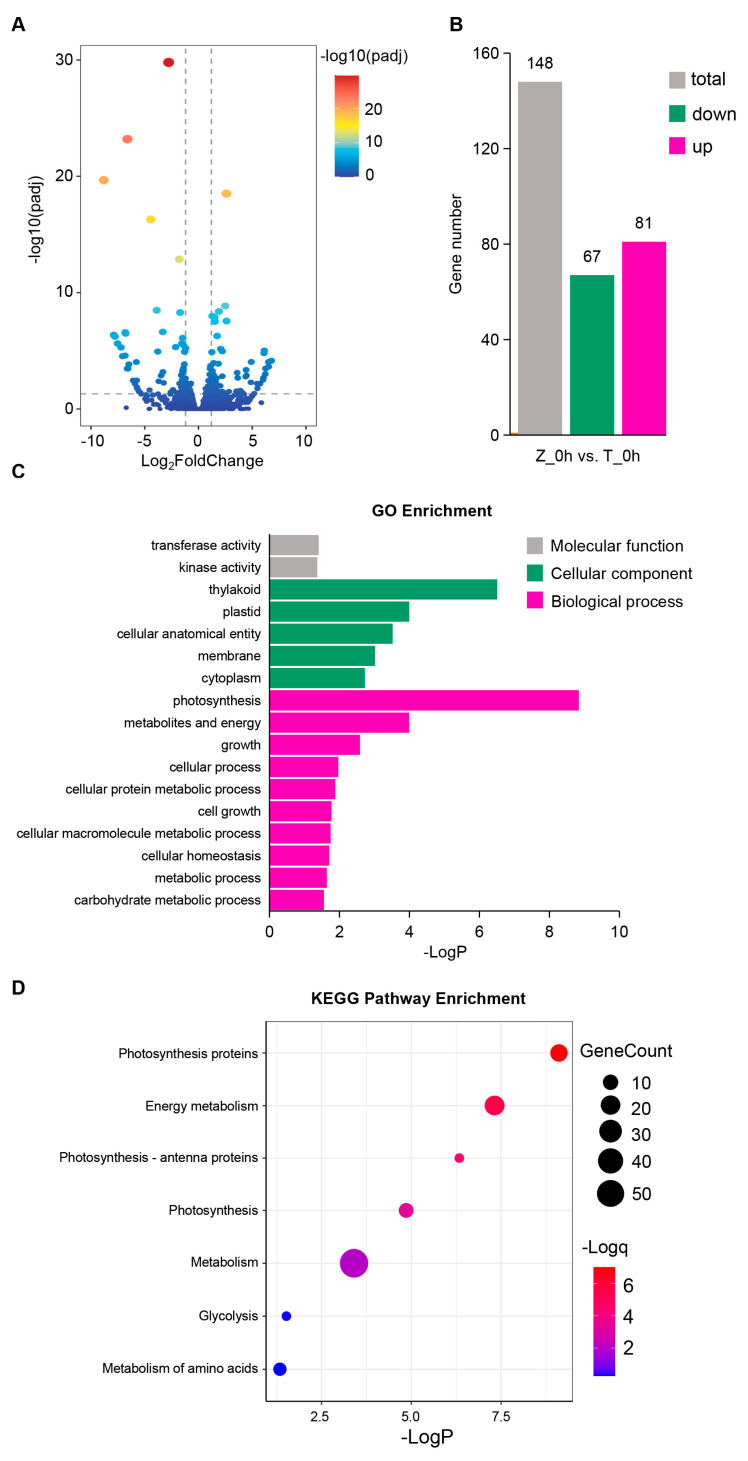
Analysis of differentially expressed genes (DEGs) between Towada and ZL31 at the 0 h cold treatment. (**A**) Volcano plot showed DEGs by ZL31 vs. Towada at 0 h (*p* < 0.01 and |Log_2_FoldChange| > 1). (**B**) The numbers of up−regulated (Magenta) and down−regulated (green) genes in ZL31 compared with Towada at 0 h cold treatment. (**C**,**D**) GO and KEGG pathway enrichment analysis of 148 DEGs. −LogP and −Logq represent the significance of GO and KEGG enrichment.

**Figure 4 ijms-24-01914-f004:**
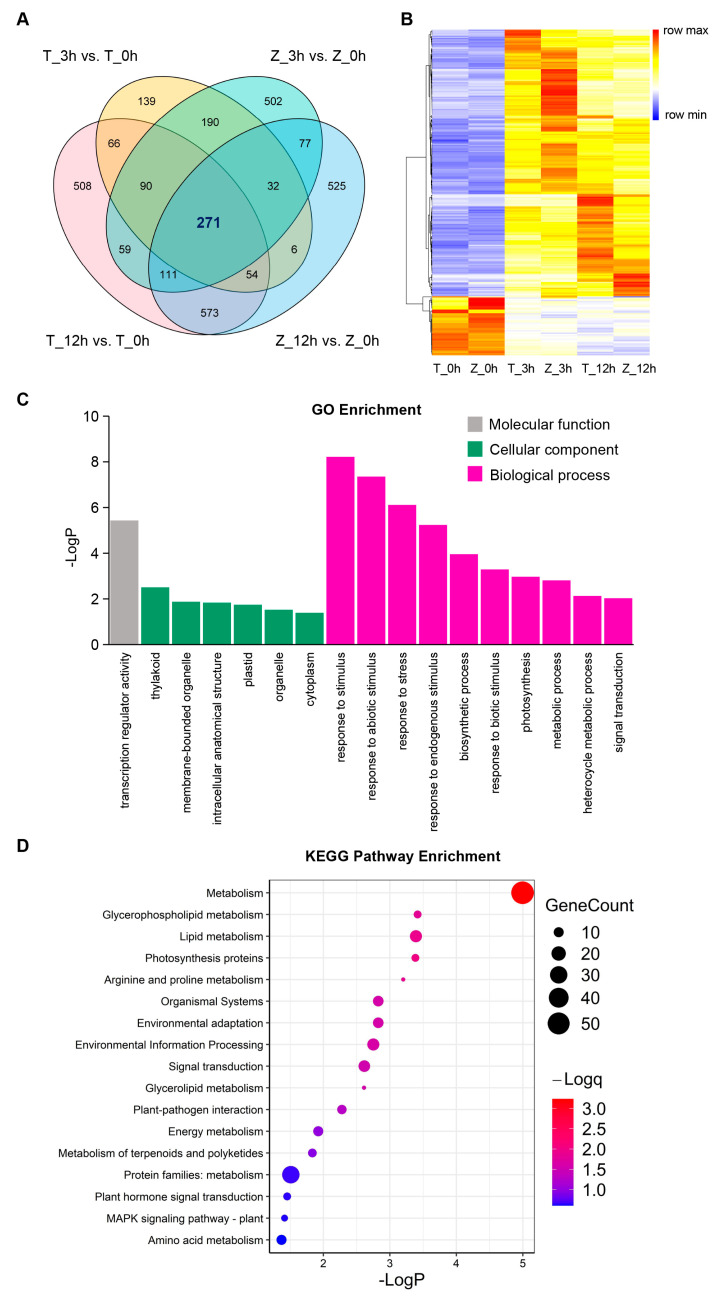
Pathway enrichment of DEGs in response to cold stress. (**A**) Venn diagram of DEGs induced by cold stress in Towada and ZL31 at the two time points. (**B**) Hierarchical clustering of the 271 DEGs based on FPKM values in Towada and ZL31. (**C**,**D**) GO and KEGG pathway enrichment analysis of the 271 DEGs.

**Figure 5 ijms-24-01914-f005:**
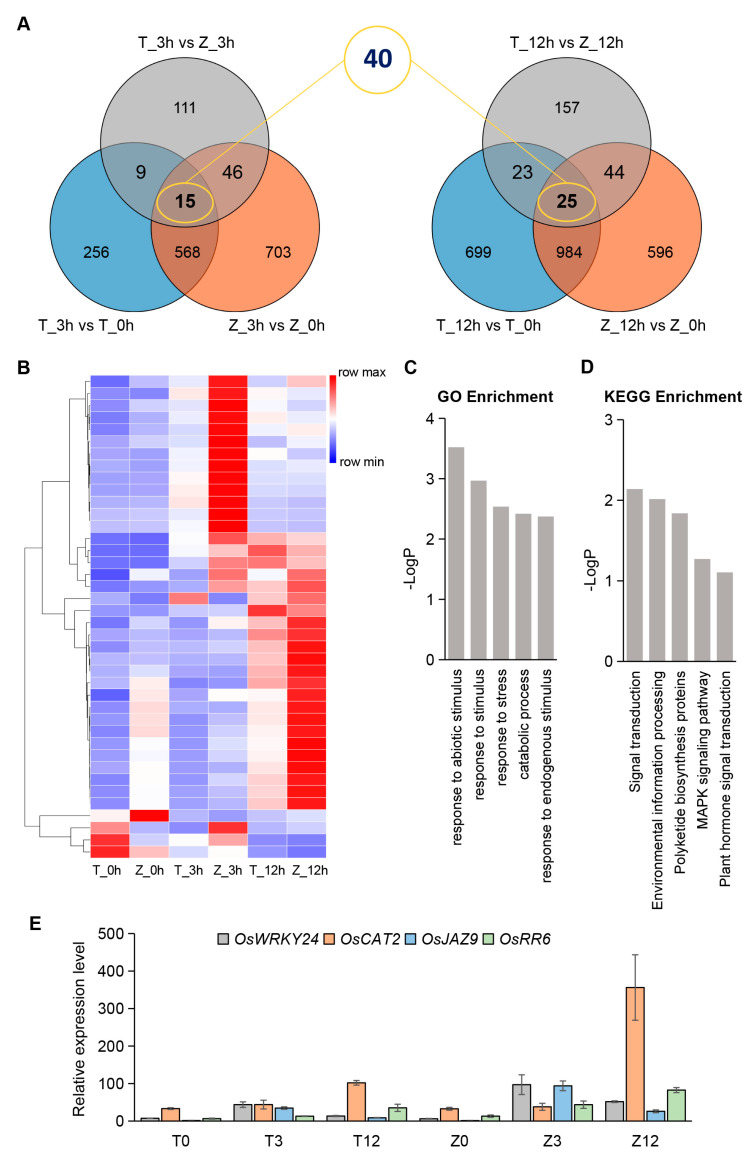
Identification and pathway analysis of core DEGs related to differential cold sensitivity between Towada and ZL31. (**A**) Venn diagram of DEGs at 3 and 12 h of cold treatment comparing Towada with ZL31. The number in the yellow circle represents the total DEGs of the two linked groups. (**B**) Hierarchical clustering of the 40 DEGs based on FPKM values in Towada and ZL31. (**C**,**D**) GO and KEGG pathway enrichment analysis of the 40 DEGs. The *y*− and *x*−axes represent the negative log_10_P value and the top 5 items with the most significant differences, respectively. (**E**) Relative expression levels of *OsWRKY24*, *OsCAT2*, *OsJAZ9,* and *OsRR6* by qRT−PCR. All assays were performed with three biological and three technical replications. Error bars, mean + SE.

**Table 1 ijms-24-01914-t001:** List of core genes, the expression spectrum, and the KEGG pathway in Towada and ZL31.

RAPD	MSU	Gene	Towada (FPKM)			ZL31 (FPKM)			KEGG Pathway
			0 h	3 h	12 h	0 h	3 h	12 h	
Os01g0826400	LOC_Os01g61080	*OsWRKY24*	1.93	13.59	7.34	2.07	38.85	9.39	Plant hormone signal transduction, environmental information processing, MAPK signaling pathway
Os02g0115700	LOC_Os02g02400	*OsCAT2*	6.26	5.70	57.10	8.86	9.97	121.59	Plant hormone signal transduction, environmental information processing, MAPK signaling pathway
Os03g0180800	LOC_Os03g08310	*OsJAZ9*	1.20	28.29	7.60	2.50	65.23	7.42	Plant hormone signal transduction, environmental information processing, plant hormone signal transduction
Os04g0673300	LOC_Os04g57720	*OsRR6*	4.09	9.76	30.81	13.34	21.52	64.58	Plant hormone signal transduction, environmental information processing, plant hormone signal transduction

**Table 2 ijms-24-01914-t002:** The sequence variations of the candidate genes between Towada and ZL31.

Gene	Chr.	Position	Nipponbare	Variation position	Towada	ZL31
LOC_Os01g61080 (*OsWRKY24*)	1	35,346,188	G	Promoter	G	GA
	1	35,346,977	G	Promoter	G	A
	1	35,347,362	A	Promoter	A	C
	1	35,347,467	T	Promoter	T	C
	1	35,347,538	T	Promoter	T	C
	1	35,347,546	A	Promoter	A	G
	1	35,347,630	T	Promoter	T	C
	1	35,347,905	CG	Promoter	CG	C
	1	35,348,004	C	5′ UTR	C	A
	1	35,348,165	T	Nonsynonymous coding	T	C
	1	35,348,568	C	Synonymous coding	C	T
	1	35,348,651	T	Nonsynonymous coding	T	C
	1	35,349,405	A	Nonsynonymous coding	A	G

## Data Availability

The raw data files of transcriptomic analysis have been uploaded to the National Center for Biotechnology Information (NCBI) under the BioProject PRJNA916930, biosamples (SAMN32599624, SAMN32599625, SAMN32599626, SAMN32599627, SAMN32599628, SAMN32599629, SAMN32599630, SAMN32599631, SAMN32599632, SAMN32599633, SAMN32599634, SAMN32599635, SAMN32599636, SAMN32599637, SAMN32599638, SAMN32599639, SAMN32599640, and SAMN32599641), and sequence read archive (SUB12496412). The raw data files used for genome analysis have been uploaded to the National Center for Biotechnology Information (NCBI) under the BioProject PRJNA916819, biosamples (SAMN32497207, SAMN32497208, SAMN32497209, and SAMN32497210), and sequence read archive (SUB12494811).

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
