# Peer review of "Transcriptome Analysis Revealed the Dynamic and Rapid Transcriptional Reprogramming Involved in Cold Stress and Related Core Genes in the Rice Seedling Stage"

_ijms, 2023, doi:10.3390/ijms24031914_

Round 1

Reviewer 1 Report

This paper compares the transcriptomics of 2 rice NILs in response to cold stress at various time points using RNA-Seq.  The authors conduct simple, yet scientifically sound studies to isolate and compare 4 genes that were differentially expressed between the two NILs.   After doing sequence alignments of genes to identify important nucleotides, it was determined that some may be important in binding to other TFs to regulate cold response where as other genes had different control mechanisms.  I find no major issues with the manuscript, but offer the following minor suggestions:

Line 21 (NIL)- please include all full names the first time you introduce an acronym.  ex. Near Isogenic Lines (NILs)

Line 100- please cite the previous study that you reference

Line 112- replace "significance of data" with "differences between means"

In all figure captions:  the letter should come before the description for example:  A) Venn Diagram. B) GO Ontologies. C) KEGG Diagram.

Figure 2A- can you circle or highlight the specific clusters from the PCA?

Figure 2B is confusing.  Might be better to have 2 separate graphs or have down regulated genes going downward from a 0 axis rather than on top of the other columns. 

Figure 3B- Explain what this graph is.  "vs" is unclear and I don't know what you are comparing (differences? % difference??)

Figure 3D- The circles for gene counts are difficult to differentiate, please use different symbols.   Same for Figure 4D

Figure 4A- This Venn Diagram, although very pretty is not clear.  These data would be better listed in a table or matrix. 

Figure 4B and 5B- Put time points side by side in the heat maps.  Ex. T_0h next to Z_0h etc. 

Line 224- Delete everything before the bolded Table 1. 

Methods Section- add spaces/lines between text sections. 

Reviewer 2 Report

The work addresses an interesting topic in cold-related transcriptional reprogramming process and provides some potential candidates involved in cold stress. However, I am concerned about several important issues of this study and thus I consider that a major revision of this manuscript needs to be done.

1Transcriptome data lack validation. Please use qRT-PCR to verify key genes in transcriptome data such as OsWRKY24, OsCAT2, OsJAZ9 and OsRR6.

2This paper mainly analyzed transcriptome data and lacked experimental verification data. Suggesting to overexpress Towada OsWRKY24 gene in rice ZL31 and observe the resistance to cold in the transgenic ZL31 rice.

3Explaining the meaning of LogP and Logq in the Figure 3 legend.

4Please list the 40 differentially expressed genes in the Figure 5 as a Supplementary material.

5Suggesting that “theoretical basis” be replaced by “new insight” in Line 322.

6The original transcriptome and genome resequencing data have not been deposited in a public database. The data should be deposited as soon as possible, and the accession number should be marked in the Data Availability Statement of the manuscript.

Round 2

Reviewer 2 Report

Accept in present form